# The Evolution of Mitochondrial Genomes between Two *Cymbidium* Sister Species: Dozens of Circular Chromosomes and the Maintenance and Deterioration of Genome Synteny

**DOI:** 10.3390/genes14040864

**Published:** 2023-04-03

**Authors:** Xiaoling Li, Mengqing Zhe, Yiwei Huang, Weishu Fan, Junbo Yang, Andan Zhu

**Affiliations:** 1Germplasm Bank of Wild Species, Yunnan Key Laboratory of Crop Wild Relatives Omics, Kunming Institute of Botany, Chinese Academy of Sciences, Kunming 650201, China; 2University of Chinese Academy of Sciences, Beijing 100049, China

**Keywords:** mitogenome, multi-chromosomes, genome synteny, recombination, nutritional modes

## Abstract

Plant mitochondrial genomes (mitogenomes) exhibit fluid genome architectures, which could lead to the rapid erosion of genome synteny over a short evolutionary time scale. Among the species-rich orchid family, the leafy *Cymbidium lancifolium* and leafless *Cymbidium macrorhizon* are sister species with remarkable differences in morphology and nutritional physiology. Although our understanding of the evolution of mitochondria is incomplete, these sister taxa are ideal for examining this subject. In this study, the complete mitogenomes of *C. lancifolium* and *C. macrorhizon*, totaling 704,244 bp and 650,751 bp, respectively, were assembled. In the 2 mitogenomes, 38 protein-coding genes, 18 *cis*- and 6 *trans*-spliced introns, and approximately 611 Kb of homologous sequences are identical; overall, they have 99.4% genome-wide similarity. Slight variations in the mitogenomes of *C. lancifolium* and *C. macrorhizon* in repeat content (21.0 Kb and 21.6 Kb, respectively) and mitochondrial DNA of plastid origin (MIPT; 38.2 Kb and 37.5 Kb, respectively) were observed. The mitogenome architectures of *C. lancifolium* and *C. macrorhizon* are complex and comprise 23 and 22 mini-circular chromosomes, respectively. Pairwise comparisons indicate that the two mitogenomes are largely syntenic, and the disparity in chromosome numbers is likely due to repeat-mediated rearrangements among different chromosomes. Notably, approximately 93.2 Kb *C. lancifolium* mitochondrial sequences lack any homology in the *C. macrorhizon* mitogenome, indicating frequent DNA gains and losses, which accounts mainly for the size variation. Our findings provide unique insights into mitogenome evolution in leafy and leafless plants of sister species and shed light on mitogenome dynamics during the transition from mixotrophy to mycoheterotrophy.

## 1. Background

Mitochondria are the centers of energy production and metabolic conversion within cells and play a vital role in plant productivity and development [1,2,3]. Plant mitochondrial genomes (mitogenomes) have been long recognized as evolving rapidly in structure and slowly in sequence [4,5]. In fact, sequence divergence among angiosperm mitogenomes is typically slow [6,7], approximately 4- and 20-fold slower than in their chloroplast and nuclear counterparts, respectively [8,9]; however, exceptions have been observed in several lineages [9,10,11,12]. The genome size of angiosperm mitogenomes can vary substantially between closely-related species or even among populations [13]. Angiosperm mitochondrial genomes typically range from 200 to 800 Kb in size, but genome sizes have exceeded 1 Mb in some species [14,15,16]. Gene order and genome structure are rarely conserved in angiosperm mitogenomes [5,6]. The rapid deterioration of genome synteny in the mitogenome is tightly associated with its high recombination rate, which is mediated mainly by large (e.g., >1 Kb) repeats [7,17,18]. The complexity of angiosperm mitogenome architectures is also substantiated by their in vivo structural diversity, including linear, circular, and branched structures [19].

With more than 28,000 species, Orchidaceae is one of the most diverse angiosperm families, accounting for approximately 9% of all vascular plants [20,21]. Within the Orchidaceae, the genus *Cymbidium* comprises about 70 species, which are distributed mainly from East and Southeast Asia to Australia, and exhibits an extraordinary diversity in lifestyles, including terrestrial, epiphytic, and lithophytic life forms [22]. The genus *Cymbidium* can be divided into three subgenera (subg.): *Cymbidium*, *Jensoa*, and *Cyperorchis* [22]. 

*C. lancifolium* Hook and *Cymbidium macrorhizon* Lindl are sister species in the Subg. *Jensoa*, which diverged from each other approximately 2.4 million years ago (Mya) [23,24,25]. The two species differ substantially in their morphological traits and nutritional modes. For example, *C. lancifolium* has leaves and is apparently a mixotrophic orchid species [26]. In contrast, *C. macrorhizon* lacks leaves, leading to increased dependence on fungal associations for nutrient acquisition [24,27].

To date, only a few mitochondrial genome sequences have been reported for the species-rich Orchidaceae family. Previous studies have indicated that the mitochondrial genome of the mycoheterotrophic orchid, *Gastrodia elata*, has markedly expanded to >1339 Kb, with many photosynthesis-related genes being lost [28]. The mitochondrial genome of the moth orchid (*Phalaenopsis aphrodite*) is approximately 576 Kb in size, with 27% and 9% of the sequence derived from the nucleus or plastid, respectively [29]. The newly reported mitochondrial genome of *Paphiopedilum micranthum* consists of 26 circular chromosomes with a total length of 447,368 bp, of which many are intracellular gene transfers [30].

In this study, we assembled the complete mitochondrial genomes of *C. lancifolium* and *C. macrorhizon* based on a combination of long- and short-read sequencing. The aims of this study include (1) a comparative analysis of mitochondrial genome architectures of leafy *C. lancifolium* and leafless *C. macrorhizon*; and (2) an investigation of the evolutionary mechanisms underlying the maintenance and deterioration of genome synteny in multi-circular molecules.

## 2. Materials and Methods

### 2.1. Plant Materials and Sequencing

Seedlings of two *Cymbidium* sister species (*C. lancifolium* and *C. macrorhizon*) were collected from the Orchid Germplasm Resource Garden of Kunming Institute of Botany and Changchong Mountain of Kunming (Yunnan, China), respectively (Appendix A). Total genomic DNA was extracted from fresh leaves of *C. lancifolium* by using a modified cetyltrimethylammonium bromide (CTAB) protocol [31]. For genome sequencing, a genomic library with 500 bp fragments was prepared for short-read sequencing in the paired-end sequencing mode on the DNBSEQ-T7 platform, and a 15-Kb insertion library was constructed for long-read sequencing on the Pacific Biosciences (PacBio) sequel II platform with HiFi mode (Appendix A). For the leafless *C. macrorhizon*, total genomic DNA and short-reads were obtained from both scapes (flowering stems) and rhizomes by using the CTAB protocol and next-generation method, as described for *C. lancifolium*. (Appendix A). Long reads were generated on the PacBio sequel II platform by using the Continuous Long Reads (CLR) mode with a 20 Kb insertion library (Appendix A).

### 2.2. Mitogenome Assembly and Evaluation

Approximately 9 Gb Illumina short reads were used to assemble the draft mitogenomes of *C. lancifolium* and *C. macrorhizon* with the “-R 20 -k 21,45,65,85,105 -P 1,000,000 -F embplant_mt” option implemented in GetOrganelle version 1.7.5 [32,33,34,35]. The assembly graphs were visualized by using Bandage based on the raw graphical fragment assembly (GFA) file produced by GetOrganelle [36], and the read depths of contigs as well as the links of graph edges were manually checked. Short (<1 Kb) contigs with low sequencing depths (<7×) or without direct links to other mitochondrial contigs were removed from further analyses.

The initial assembly outputs for *C. lancifolium* and *C. macrorhizon* contained 363 contigs and 228 contigs, with a total length of 843,278 bp and 787,850 bp, respectively (Appendix A). Then, Burrows–Wheeler Aligner (BWA)-mem [37] was used to identify putative mitochondrial DNA-derived PacBio long reads from approximately 14 Gb raw long reads by mapping to these draft assemblies. Canu v.2.2 [38] was used to assemble the mitochondrial genome of *C. lancifolium* and *C. macrorhizon* independently, based on the -pacbio-hifi and -pacbio modes, respectively. Ultimately, two complete mitochondrial genomes with multiple circular molecules were assembled for the two *Cymbidium* species. Pilon version 1.24 [39] was used to correct the mitogenome of *C. macrorhizon*, which was assembled by using long reads generated via the PacBio CLR mode.

To verify the integrity of the mitogenome structure of *C. lancifolium* and *C. macrorhizon*, the sequenced long reads were mapped to their corresponding assemblies by using map-hifi and map-pb modes of minimap2 v.2.24 [40], index files were generated with SAMtools and check GAP via the integrative genomics viewer (IGV) [41,42], and the average read depth of each mitogenome was estimated with BEDTools v.2.30.0 [40,42]. To avoid mapping bias at the ends of each chromosome, we also redefined a new starting point using the command “restart -i 5000” implemented in SeqKit [43] to check the read coverage. For a final test, we mapped the initial draft mitochondrial assembles (based on Illumina short reads) to the complete mitogenomes (based on PacBio long reads) by using the asm10 mode of Minimap2. The mapping rates of *C. lancifolium* and *C. macrorhizon* contigs were 98.38% and 96.26%, respectively, giving strong cross-validation for the completeness of PacBio-based mitogenome assemblies.

### 2.3. Genome Annotation

The mitogenomes were annotated by using the “Live annotate & predict” tool in Geneious R7 [44]. Two mitochondrial genome references, *Arabidopsis thaliana* (NC_037304) and *Nymphaea colorata* (NC_037468), were downloaded from GenBank. Gene structure was manually checked to refine the in-frame start and stop codons and adjust the exon-intron boundaries. tRNAscan-SE [45] was used to identify mitochondrial tRNAs with default parameters. The mitogenome maps were drawn by using TBtools [46]. The assembled mitogenomes and their annotations were deposited in GenBank under the accession numbers OQ024442-OQ024464 and OQ029542-OQ029563.

### 2.4. Identification of Repeats and Mitochondrial DNA of Plastid Origin

Simple sequence repeats (SSR) were analyzed by using MISA (2017, https://webblast.ipk-gatersleben.de/misa/ accessed on 20 March 2023) with the following parameters: (1/10) (2/5) (3/4) (4/4) (5/3) (6/3). Disperse repeats within and across chromosomes of each mitogenome were identified by using the basic local alignment search tool (BLAST)N v.2.12.0 to search against themselves with a minimum alignment length of 100 bp and minimal identity of 95%. The chloroplast genomes of *C*. *lancifolium* and *C*. *macrorhizon* were downloaded from the National Center for Biotechnology Information (NCBI) (MW582681 and MW582687). The plastid-derived sequences (mitochondrial DNA of plastid origin) were identified by mapping the *C. lancifolium* and *C. macrorhizon* plastomes to their corresponding mitogenomes with BLASTN by using the same criterion as for the disperse repeat analysis.

### 2.5. Genome Similarity and Synteny Analysis

The whole-genome average nucleotide identity between *C. lancifolium* and *C. macrorhizon* mitogenomes was calculated by using FASTANI [47] with default parameters. Whole-mitogenome alignment was performed by using BLASTN to identify homologous chromosomes. Based on the alignment results, we used SeqKit to reset the start site and manually sort the results, and Mauve v.2.4.0 [48] was used to determine and visualize genome synteny as locally collinear blocks with a minimal length of 1 Kb.

### 2.6. Identification and Verification of Species-Specific Mitochondrial Sequences

Homologous sequences were identified by using BLASTN searches against the *C. lancifolium* and *C. macrorhizon* mitogenomes with sequence identity >98% and a minimum alignment length of 100 bp. Species-specific mitochondrial sequences were identified by parsing the BLASTN alignment results. The evolution of mitochondrial sequence composition was measured by the maintenance and turnover of homologous sequences [49]. Moreover, homologous sequences were extracted from syntenic blocks and realigned with Muscle [50] to identify insertions/deletions (indels). Six large (each > 1 Kb in size) indels that were caused by the maintenance of *C. lancifolium*-specific sequences were identified, and these indels mostly accounted for the mitogenome size difference between *C. lancifolium* and *C. macrorhizon*. We validated these large indels by mapping *C. macrorhizon* short reads to the *C. lancifolium* mitogenome by using BWA and visualized them by using IGV [37,41,42].

## 3. Results

### 3.1. The Mitogenomes of Sister Species C. lancifolium and C. macrorhizon Comprise Dozens of Minicircular Chromosomes

De novo assemblies using PacBio long reads generated two complete mitogenomes with multiple circular molecules (Figure 1; Table 1). The mitogenome of the leafy *C. lancifolium* comprises 23 circular chromosomes that vary from 23,637 to 46,518 bp in size, totaling 704,244 bp (Appendix A). The leafless *C. macrorhizon* mitogenome has a total length of 650,751 bp and consists of 22 circular chromosomes, ranging from 20,861 to 46,822 bp in size. Both the *C. lancifolium* and *C. macrorhizon* mitogenomes exhibit a 44% GC content, similar to other angiosperms [51]. Long reads were mapped back to their corresponding mitogenome to verify genome continuity, and the average depth of assembled mitogenomes was ~110× (Appendix A).

### 3.2. Maintenance of Gene and Intron Content in the C. lancifolium and C. macrorhizon Mitogenomes

The *C. lancifolium* and *C. macrorhizon* mitogenomes are nearly identical regarding gene and intron content (Table 2 and Figure 1). A total of 38 unique protein-coding genes and 3 rRNAs was identified in each mitogenome. In total, 8 *cis*- and 6 *trans*-spliced introns were found in 10 mitochondrial genes (*nad1*, *nad2*, *nad4*, *nad5*, *nad7*, *cox2*, *ccmFC*, *rpl2*, *rps3*, and *rps10*). Notably, *mttB* lacks the standard start codon (ATG), which is consistent with previous findings [9,52]. Furthermore, *nad1* likely starts with ACG but could be converted to AUG at the transcript level through RNA editing. In the mitogenomes of *C. lancifolium* and *C. macrorhizon*, 21 and 19 tRNAs, respectively, were annotated.

### 3.3. Repeat Content and Mitochondrial DNA of Plastid Origin (MIPT)

A total of 130 and 128 tandem repeats (TRs) was detected in the mitochondrial genomes of *C. lancifolium* and *C. macrorhizon* (Appendix A), respectively; mononucleotide TRS (~46%) were the most common, followed by trinucleotide (~27%) and dinucleotide (~22%) TRs. Dispersed repeats, ranging from 107 bp to 1338 bp, are prevalent, accounting for approximately 2.97% and 3.34% of the *C. lancifolium* and *C. macrorhizon* mitogenomes (Table 1), respectively. Intriguingly, nearly 65% of dispersed repeats are remarkably similar in sequence within each species. These repeats can be categorized into two distinct classes (group A and group B) based on their conserved regions (CRs). Group A contains 18 repetitive sequences from 18 chromosomes of the *C. lancifolium* mitogenome that share an identical 212 bp CR (Appendix A) and 19 repetitive sequences from 19 chromosomes in *C. macrorhizon* that share an identical 140 bp CR (Appendix A). In group B, the *C. lancifolium* and *C. macrorhizon* mitogenomes have 6 repeats that share conserved regions of 149 and 148 bp, respectively (Appendix A).

Large chloroplast genome segments have transferred into the mitogenomes of the two *Cymbidium* species. A total of 38,185 bp and 37,518 bp MIPT fragments was identified in the mitogenomes of *C. lancifolium* and *C. macrorhizon*, respectively, totaling approximately 5–6% of the genome sizes (Table 1, Appendix A). The identified MIPT fragments vary substantially in length between the *C. lancifolium* and *C. macrorhizon* mitogenomes, with the two largest fragments being 9.4 Kb and 11.6 Kb, respectively. The insertion of these MIPT fragments into these circular chromosomes is not uniformly distributed. For example, 3 MIPT fragments, which were 15.0 Kb in total length, with the largest insertion being 11.6 Kb, were identified in *C. macrorhizon* Chr01 (chromosome size = 46.8 Kb), and a total of 12.7 Kb MIPT sequences was identified in *C. lancifolium* Chr02 (chromosome size = 45.7 Kb). In contrast, there are 13 and 11 chromosomes of *C. lancifolium* and *C. macrorhizon* mitogenomes, respectively, without any MIPT insertions.

### 3.4. Sequence and Genome Synteny Evolution between the C. lancifolium and C. macrorhizon Mitogenomes

The whole-mitogenome average nucleotide identity is 99.4% between *C. lancifolium* and *C. macrorhizon*, which is consistent with their sister species relationship and relatively recent divergence time (approximately 2.4 Mya) [24]. Homologous sequences of approximately 611 Kb were shared in both species (Figure 2A), covering 86.8% of the *C. lancifolium* mitogenome and 93.9% of the *C. macrorhizon* mitogenome. However, the variation in mitogenome sizes was mainly due to the deletion of several large DNA segments in *C. macrorhizon* and/or DNA gains in *C. lancifolium*. For example, approximately 48 Kb mitochondrial sequences from five *C. lancifolium* chromosomes (Chr01, Chr04, Chr06, Chr10, and Chr12) lacked any homology in *C. macrorhizon* (Figure 2B). These non-homologous sequences were nested in other homologous sequences, leading to insertions and deletions (indels) during genome alignment. To verify this, we examined all 6 large (>1 Kb) indels, ranging from 3.2 to 19.8 Kb. All six of these regions are located in intergenic regions and are continuous in both mitogenomes. As expected, inter-specific mapping of the *C. macrorhizon* reads to the *C. lancifolium* mitogenome revealed that these indel regions were less covered by reads (Figure 2B).

The mitogenome architectures of *C. lancifolium* and *C. macrorhizon* are generally consistent at the macrosynteny level (Figure 3). Specifically, 20 out of the 22 total chromosomes in *C. macrorhizon* had a 1:1 homologous relationship with individual chromosome in *C. lancifolium* (Appendix A). Despite the largely maintained genome synteny between these 2 sister species, a total of 14 rearrangements has occurred between the *C. lancifolium* and *C. macrorhizon* mitogenomes. The most notable genomic rearrangements were found in Chr07 and Chr08. Chr08 in *C. macrorhizon* was homologous to parts of Chr10 and Chr23 in *C. lancifolium*, while Chr07 has a tripartite network of homologous relationships to partial Chr01, Chr04, and Chr23 sequences in *C. lancifolium* (Figure 3B).

## 4. Discussion

Plant mitochondrial DNAs (mtDNAs) exhibit striking structural diversity within and among species, with architectures that can shift rapidly between linear, circular, and branched structures in vivo [19,53,54]. This study revealed multi-chromosomal structures of mitogenomes within two *Cymbidium* sister species, each consisting of >20 mini-circular chromosomes (20–47 Kb; Figure 1). Multi-chromosomal mitogenomes have independently evolved in several lineages, such as *Cucumis* [55], *Silene* [12], *Ajuga* [56], and a parasitic plant by the name of *Rhopalocnemis phalloides* [57]. Notably, for these multi-chromosomal mitogenomes, both the chromosome numbers and their sizes can differ substantially in closely related species and even among populations. For example, the mitogenome of *Silene noctiflora* exhibits differences in chromosome numbers (59 and 63) and sizes (6.73 Mb and 7.14 Mb) between accessions [12,58]. Rapid gain or loss of several entire chromosomes has been suggested as a crucial evolutionary mechanism accounting for these disparities [58].

Our syntenic analysis of the mitogenomes of the sister species, *C. lancifolium* and *C. macrorhizon*, provided an alternative mechanism for differences in chromosome number and size. Chromosome-scale syntenic relationships could rapidly erode via inter-chromosomal rearrangements, resulting in disparity of chromosomal numbers (Figure 3B). Inter-chromosomal rearrangements are presumably mediated by recombinationally-active repeated sequences, which are widespread in vascular mitogenomes [54,59,60]. We found that an identical large (1256 bp) repeat pair is shared by Chr07 and Chr08 in the *C. macrorhizon* mitogenome, suggesting that these chromosomes might be recombination hotspots.

Intriguingly, two types of conserved sequences that are shared by nearly all chromosomes were dominant among the identified dispersed repeats (Appendix A). It is unclear why and how such repeats have been maintained or accumulated in plant mitogenomes. A recent study in the parasitic plant *R. phalloides* has identified an 896 bp conserved region that is shared by all chromosomes and could act as the origin of replication by forming stem–loop secondary structures [57]. If this mitochondrial replication mechanism is not unique to parasitic plants, it is reasonable to propose that similar adaptive roles might affect the maintenance of these accumulated repeats.

The leafless *C. macrorhizon* differs considerably in morphology and nutritional mode from its sister species *C. lancifolium* (Appendix A). However, the mitogenomes of these two species are generally consistent in sequence composition (gene and intron content) and macrosynteny (except for the few rearranged chromosomes). This result mirrors our findings of minor genetic differences between the chloroplast genomes of these two morphologically distinct sister species [61]. The evolution of the genomes of *C. macrorhizon* organelles contrasts sharply with the findings for a fully mycoheterotrophic plant, *G. elata*, which has a dramatic reduction in gene content of its organelle genomes [28]. This difference might suggest that *C. macrorhizon* is at an early stage of a transition to a fully mycoheterotrophic lifestyle.

The mitogenomes of *C. lancifolium* and *C. macrorhizon* are conserved in terms of sequence organization. The multi-chromosome structure is consistent with the recently reported mitogenome of *P. micranthum*, which consists of 26 circular chromosomes, ranging in size from 5973 bp to 32,281 bp [30]. For other orchid species, it has been reported that the mitogenome is fragmented, suggesting that the multi-circular structure of the mitogenome is likely common in species within the orchid family; plastid-derived DNA fragments represented 5% and 6% of the *C. lancifolium* and *C. macrorhizon* mitogenome, respectively, with similar findings reported for *P. aphrodite* (9%) and *P. micranthum* (10%) [29,30]. Frequent inter-chromosomal recombination allows the mitochondrial genome to easily become a receptor for the lateral transfer of foreign genomes, which is consistent with the large number of recombination events that have occurred in the mitogenomes of different orchid species.

## 5. Conclusions

We accurately assembled the mitogenomes of *C. lancifolium* and *C. macrorhizon* by using a combination of short-read and long-read data. Both mitogenomes have multi-circular architectures, which are composed of more than 20 sub-genomes. Furthermore, a large number of fragments have been transferred from the chloroplast genome, reflecting the complex and dynamic evolution of the mitogenome of orchid species. The protein-coding genes in both species are consistent, and the mitogenome-wide average nucleic acid identity is 99.4%, with more than 86% homologous sequences, which is consistent with the evolutionary relationship of these sister species. However, species-specific sequences are still present, and we identified the insertion/deletion of several large fragments, which directly led to the difference in mitogenome size. In addition, we suggest that the difference in the number of chromosomes resulted from intermolecular or intramolecular rearrangements. Interestingly, we identified two sets of homologous repeats that appeared 6–19 times in the mitochondrial genomes of both species. However, while the reasons for the large accumulation and maintenance of such repeats by species are currently unclear, we proposed an adaptive role for these homologous repeats. The multi-circle structure of the mitochondrial genome of Orchidaceae is not the first to be reported, and in addition to the current fragmented mitochondrial genome in Orchidaceae, we speculate that multi-circle structures may be common in the mitochondrial genome of orchid species. Information on the mitochondrial genome of orchid species is largely unavailable. Further studies are needed to unravel the evolution of the mitogenome of orchids.

## Figures and Tables

**Figure 1 genes-14-00864-f001:**
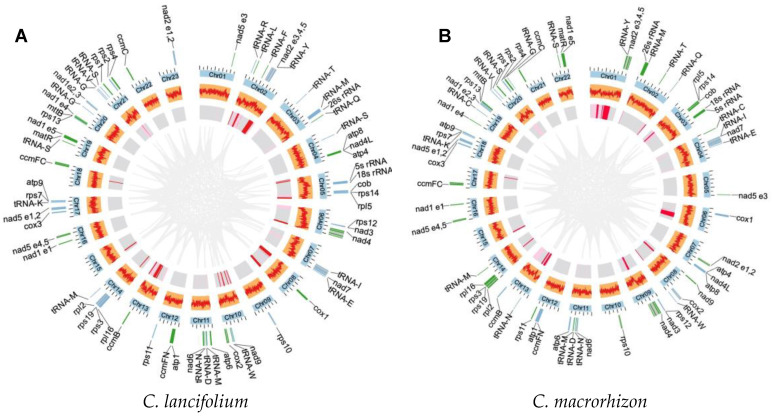
The complete mitogenomes of *C. lancifolium* and *C. macrorhizon*. (**A**) The mitogenome structure and features of the leafy *C. lancifolium*. (**B**) The mitogenome structure and features of the leafless *C. macrorhizon.* From outermost to innermost, circle I: mitochondrial genes transcribed clockwise and counterclockwise are shown in green and blue, respectively; circle II: mitochondrial chromosomes; circle III: GC content; circle IV: MIPT sequences; circle V: repeats across chromosomes.

**Figure 2 genes-14-00864-f002:**
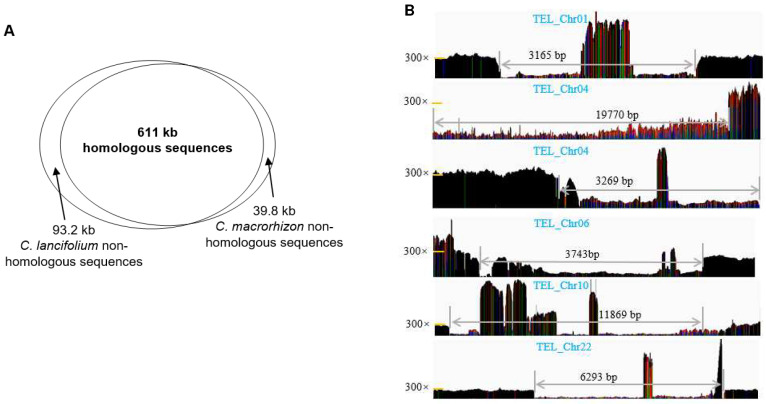
The maintenance and turnover of homologous sequences in the *C. lancifolium* and *C. macrorhizon* mitogenomes. (**A**) The relative composition of homologous and non-homologous sequences in the two mitogenomes. (**B**) Validation of *C. lancifolium* non-homologous sequences by using *C. macrorhizon* short reads. The gray arrows highlight the six largest indels caused by the maintenance of *C. lancifolium*-specific sequences.

**Figure 3 genes-14-00864-f003:**
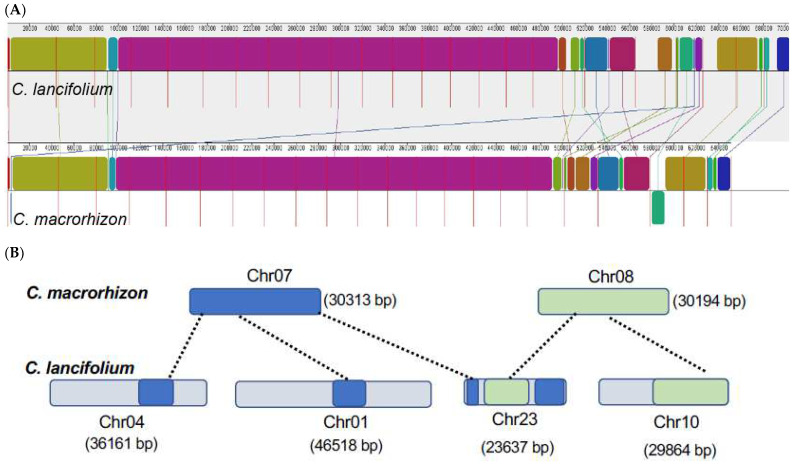
The maintenance and turnover of synteny between the *C. lancifolium* and *C. macrorhizon* mitogenomes. (**A**) The overall whole-genome synteny between the two mitogenomes. Syntenic blocks were identified by using progressiveMauve implemented in Mauve. Upper: *C. lancifolium*; lower: *C. macrorhizon.* (**B**) Inter-chromosomal synteny relationship between the *C. lancifolium* and *C. macrorhizon* mitogenomes.

**Table 1 genes-14-00864-t001:** General genome features of the assembled *C. lancifolium* and *C. macrorhizon* mitogenomes.

	Size (bp)	Chromosomes	Protein-Coding Genes	MIPT (bp)	Repeats (bp)
*C. lancifolium*	704,244	23	38	53,587	20,951
*C. macrorhizon*	650,751	22	38	50,901	21,619

**Table 2 genes-14-00864-t002:** Genes encoded in the *C. lancifolium* and *C. macrorhizon* mitogenomes. * indicates mitochondrial genes with introns.

Category	Gene	
Complex I	nad1 * nad2 * nad3 nad4 * nad4L nad5 * nad6 nad7 * nad9	
Complex III	cob	
Complex IV	cox1 cox2* cox3	
Complex V	atp1 atp4 atp6 atp8 atp9	
Cytochrome c biogenesis	ccmB ccmC ccmFN ccmFC *	
Ribosome large subunits	rpl2 * rpl5 rpl16	
Ribosome small subunits	rps1 rps2 rps3 * rps4 rps7 rps10 * rps11 rps12 rps13 rps14 rps19	
Others	matR mttB	
rRNA genes	rrn5 rrn18 rrn26	
tRNA genes	trnL-TAG trnF-GAA trnY-GTA trnT-TGT trnQ-TTG trnM-CAT trnT-TGT trnS-AGA trnE-TTG trnI-TAT trnW-CCA trnD-GTC trnN-GTT trnM-CAT trnM-CAT trnK-TTT trnS-GCT trnC-GCA trnS-TAC trnV-TAC trnG-TCC	trnY-GTA trnM-CAT trnT-TGT trnC-GCA trnE-TTC trnI-TAT trnW-CCA trnN-GTT trnM-CAT trnD-GTC trnN-GTT trnM-CAT trnK-TTT trnC-GCA trnG-TCC trnS-GCT trnV-TAC trnS-GCT trnQ-TTG

## Data Availability

All the assembled and annotated mitogenomes were submitted to GenBank under accessions OQ024442-OQ024464 and OQ029542-OQ029563.

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
