# Peer review of "The Evolution of Mitochondrial Genomes between Two Cymbidium Sister Species: Dozens of Circular Chromosomes and the Maintenance and Deterioration of Genome Synteny"

_genes, 2023, doi:10.3390/genes14040864_

Round 1

Reviewer 1 Report

As a result of the review of the manuscript entitled “The evolution of mitochondrial genomes between two Cymbidium sister species with distinct nutritional modes: dozens of circular chromosomes and the maintenance and deterioration of genome synteny”, it has been seen that it is generally clean in terms of writing language and grammar.

Below are my opinions and suggestions for specific sections.

Abstract: The abstract section seems to be sufficient in terms of the importance of the subject, methodology, results, and interpretation of the results.

Introduction: Overall the introduction section is well organized and summarizes the topic of interest. Some minor issues need to be addressed

·         Some parentheses do not have a space with the word. They look attached.

·         “The genus Cymbidium could be divided into three subgenres: Subgen. Cymbidium, Subgen. Jensoa and Subgen. Cyperorchis” can be shortened as “The genus Cymbidium could be divided into three subgenres: Cymbidium, Jensoa and Cyperorchis”

·         Lines 62-67: this paragraph seems to be written in a different style.

·         “To date, only a few draft but no complete mitochondrial genome sequences have been reported in the species-rich Orchidaceae family.” This is a bold argument, there would be some recent reports on other Orchid species. I would suggest authors to check the literature one more time. After a brief search, I found the below report, so there might be some other reports if so they need to be given in the introduction and discussed.

Yang, J.-X.; Dierckxsens, N.; Bai, M.-Z.; Guo, Y.-Y. Multichromosomal Mitochondrial Genome of Paphiopedilum micranthum: Compact and Fragmented Genome, and Rampant Intracellular Gene Transfer. Int. J. Mol. Sci. 202324, 3976. https://doi.org/10.3390/ijms24043976

·         Although the authors say that there is no complete mitogenome study in Orchidaceae, there are certainly similar and other species. And this should be given in the introduction.

·         Line 61-68 seems different in style. please check it.

M&M section:

·         Please give detail about how DNA is extracted and detailed processes.

·         What is the insert size for long and short-read sequencing?

Results

·         Some paragraphs have references. It is not common to cite references in the results section if it is a separate section from the discussion.

·         Lines 148-149: single sentence does not look good if it is not a subtitle.

·         Lines 193-194: would not this sentence be given in the material and methods section?

Discussion

·         The discussion is written well highlighting the main outcomes of the study. But it is a bit short compared to the data obtained. It can be extended with related species and other recent similar reports.

Supplementary table and figures:

·         S. Table 1: What are DGL3-3 and TEL228? Please explain and indicate in the MS if necessary.

Overall, I would suggest a minor revision and will be happy to re-evaluate the revised version.

Regards

Author Response

Dear  reviewer,

Thank you very much for your professional comments. After careful consideration, we have updated the manuscript in response to the comments, Please see the attachment.

Reviewer 2 Report

The authors must seek the help of a scientific editor to help them produce a viable manuscript for publication. In addition, there is a need to provide a more nuanced introduction and justification of the study, with clear objectives.  The materials and methods are sketchy and do not meet the normal standards of repeatability i.e., providing such detail and clarity that other researchers can repeat the study and validate the results of this study or otherwise.  The presentation of the results and their discussion are similarly obtuse and require major revision.

  Experimental section:. A more succinic yet complete writing should be done. Moreover the author state that a statistical analysis has been made. I believe that the authors should give more details about the analysis performed,.

Author Response

 thank you very much for your professional comments,Based on these comments and suggestions, we have made careful modifications to the original manuscript,Our summarized changes is described in red below each comment.

The authors must seek the help of a scientific editor to help them produce a viable manuscript for publication. In addition, there is a need to provide a more nuanced introduction and justification of the study, with clear objectives.  The materials and methods are sketchy and do not meet the normal standards of repeatability i.e., providing such detail and clarity that other researchers can repeat the study and validate the results of this study or otherwise.  The presentation of the results and their discussion are similarly obtuse and require major revision.

 Response: Thank you very much for your review, and we have made the following improvements according to your suggestions:

    1. We have carefully proofread the article and corrected the formatting issues;

    2. We've added detailed details to the M&M section to ensure that readers can repeat this research;

    3. We have extended the discussion section by comparing our results with that of related species.

Experimental section:. A more succinic yet complete writing should be done. Moreover the author state that a statistical analysis has been made. I believe that the authors should give more details about the analysis performed.

Response: We asked a professional editor to improve the English, logic, and grammar. No specific statistics were included in this manuscript.

Reviewer 3 Report

I found much of value in the current manuscript.   The few comments are appended to the manuscript file is attached.

A conclusion paragraph by the end manuscript is necessary. The conclusion paragraph should summarize the key ideas are discussed throughout the work, and display the final understanding on the main idea.

Author Response

 Thank you very much for your comments. We have revised the manuscript ,and added a conclusion at the end of the article, summarizing the key points of the article and our understanding of it.

Round 2

Reviewer 2 Report

I am pleased to inform you that I have thoroughly reviewed the revised version of the manuscript and believe that the authors have addressed the referees' comments satisfactorily. The manuscript is now in excellent shape for publication in Genes.